# Coagulative Nucleation in the Copolymerization of Methyl Methacrylate–Butyl Acrylate under Monomer-Starved Conditions

**DOI:** 10.3390/polym15071628

**Published:** 2023-03-24

**Authors:** Sujey G. Castellanos, V. Vladimir A. Fernández-Escamilla, Miguel Á. Corona-Rivera, Karla J. González-Iñiguez, Arturo Barrera, Francisco J. Moscoso-Sánchez, Edgar B. Figueroa-Ochoa, Israel Ceja, Martín Rabelero, Jacobo Aguilar

**Affiliations:** 1Departamento de Ciencias Tecnológicas, Centro Universitario de la Ciénega, Universidad de Guadalajara, Av. Universidad No. 1115, Col. Lindavista, Ocotlán 47819, Mexico; 2Ingeniería Química, Coordinación Académica Región Altiplano (COARA), Universidad Autónoma de San Luis Potosí, Carretera a Cedral Km 5+600, San Jose de las Trojes, Matehuala 78700, Mexico; 3Departamento de Química, Centro Universitario de Ciencias Exactas en Ingenierías, Universidad de Guadalajara, Blvd. Gral. Marcelino García Barragán No. 1421, Col. Olímpica, Guadalajara 44430, Mexico; 4Departamento de Física, Centro Universitario de Ciencias Exactas en Ingenierías, Universidad de Guadalajara, Blvd. Gral. Marcelino García Barragán No. 1421, Col. Olímpica, Guadalajara 44430, Mexico; 5Departamento de Ingeniería Química, Centro Universitario de Ciencias Exactas en Ingenierías, Universidad de Guadalajara, Blvd. Gral. Marcelino García Barragán No. 1421, Col. Olímpica, Guadalajara 44430, Mexico

**Keywords:** monomer-starved conditions, homogeneous nucleation, coagulative nucleation, methyl methacrylate, butyl acrylate

## Abstract

Coagulative nucleation in the copolymerization of methyl methacrylate–butyl acrylate (MMA-BA) via semicontinuous emulsion heterophase polymerization (SEHP) under monomer-starved conditions in latexes with high solid content (50.0 wt %) and low concentrations of surfactant is reported. The SEHP technique allows the obtention of latex with high colloidal stability and has potential industrial application in polymer synthesis. High instantaneous conversions (>90%) and a high-ratio polymerization rate/addition rate (*R_p_*/*R_a_*) ≥ 0.9 were obtained at low times until the final copolymerization, which confirmed the starved conditions in the systems at the highest surfactant concentrations. The particle size exhibited a linear size increment at conversions between 0 and 40% induced by homogeneous nucleation, a transition region between 40 and 50%, and non-linear behavior at higher conversions by coagulative nucleation. These three behaviors were also observed in the particle surfactant coverage area (*Sc*), Z-potential, particle coagulation rate (*dN_p_*/*dt*) by the Smoluchowski model, final particle size (*Dp_z_*), and number particle (*N_p_*) through the reaction. By means of transmission electron microscopy (TEM) images, the onset of coagulation was observed from 50% of conversion until the end of the reaction. In addition, in both processes of copolymerization, tacticity was displayed (mainly syndiotacticity).

## 1. Introduction

Polymeric particle formation mechanisms have been widely studied in emulsion-polymerization-modifying reaction factors such as the surfactant and initiator concentration [1,2,3], solid content [4,5], and monomer addition rate [2,6,7,8]. Several models have been proposed for the particle formation during the nucleation stage such as homogeneous nucleation, micellar nucleation, and coagulative nucleation. The synthesis mechanism can be tailored to a particular polymeric reaction system by adjusting the reaction factors enlisted previously. The semicontinuous emulsion heterophase polymerization (SEHP) process under monomer-starved conditions involves the continuously controlled feeding of the monomer or the comonomer into a monomer-free surfactant and initiator aqueous solution. In this polymerization process, monomer molecules are first in contact with the continuous medium, which is rich in radicals that produce oligomeric chains once in contact with monomer molecules. When oligomers gain sufficient weight, they are quickly stabilized by surfactant molecules to form new particles. The already-formed polymer–monomer particles continue to grow until there are no more reacted radicals or oligomers [9]. Otherwise, when the monomer is added at the same rate applied in a system under starved conditions but a low surfactant concentration is present, particle formation can occur under flooded conditions because there is not sufficient surfactant or micelles to encapsulate the oligomers [10]. SEHP yields high-solid-content latexes with a small particle size by using low surfactant content, while the batch emulsion polymerization (BEP) process produces large particles due to the usual high monomer concentrations from an early stage of the polymerization reaction [1,4,6,11]. Furthermore, the use of semicontinuous emulsion polymerization under monomer-starved conditions allows for a controlled particle size along the reaction time via a semicontinuous monomer addition [2,5,8,12,13]. The gradual arrival of the monomer at the continuous medium favors homogeneous particle nucleation due to the use of the low surfactant content in the monomer-free initial system. The direct consequence is the increase in the particle number and polymerization rate [2,13]. The analyzed data of particle size distributions (PSDs) after the cessation of the nucleation stage comprise another technique to determine evidence for the nucleation mechanism [3,14]. Feeney et al. [14] provided experimental and theoretical evidence for the manner in which an increase in the particle number positively skewed in the early reaction stage can be described by a coagulation step.

On the other hand, the nucleation process during polymerization is difficult to explain when monomers with different water solubilities appear to be copolymerized. Some investigations concluded that the monomer with higher solubility in the continuous media governs the homogeneous nucleation; therefore, it has a special influence on the particle size, particle number, and morphology in a typical emulsion polymerization process [15,16,17,18]. In the kinetics of the emulsion polymerization of methyl methacrylate (MMA), it was demonstrated that initiation occurred in the aqueous phase followed by the formation of new latex particles via the coagulation of polymer chains in the course of polymerization [19]. This nucleation mechanism can be influenced by the moderate solubility of MMA [20,21,22]. However, the polymerization mechanism changes when the monomers have low water solubility, such as in the case of butyl acrylate (BA), in which it moves into micelles at short polymerization times until an oligomeric radical entry into the monomer-swollen micelles to begin the nucleation of the particle (micellar nucleation) [20,23,24,25]. The polymerization technique has a special influence on the nucleation mechanism and thus the particle size effects. The absence of surfactant during the emulsion polymerization of MMA-BA was studied by Lee et al. in a seeded emulsion polymerization [19]. The weight ratio of MMA-BA and the effect caused in the average molecular weight was analyzed. The two-stage monomer addition applied led to a core–shell structure in the polymer particles. Sommer et al. employed atomic force microscopy (AFM) to analyze the surface morphology effects of polymer particles of PMMA/PBA in a seeded emulsion polymerization [20]. Based on AFM images, these authors suggested that a coagulation mechanism can influence the nucleation of PMMA. Ouzineb et al. [23] elucidated the nucleation mechanism above and below the critical micelle concentration (CMC) for the emulsion copolymerization of MMA-BA. Above CMC, the particle number increased throughout the polymerization (assuming that homogeneous nucleation dominated). Below CMC, there was little free surfactant to stabilize the nucleated particles, then small particles were introduced into larger particles to maintain a constant number of particles [23].

Semibatch emulsion copolymerization with high solid content exerts an important effect on the particle size and number. Chern and Hsu [24] established how the combination of an anionic surfactant with a non-ionic surfactant modifies the nucleation mechanism and particle size. In addition, these authors demonstrated that particle size had more influence by means of the initial surfactant content in the reactor charge and was independent of the initiator concentration; a coagulative nucleation mechanism occurred when the surfactant concentration was far below its CMC [24]. In copolymerized systems when the particle formation of polymers occurs under monomer-starved conditions, the process is surrounded by all of the environmental polymerized factors. Commonly, the monomer addition rate, surfactant, and polymer content are just some of the factors that influence the particle-formation mechanism and the final structure of the polymer nanoparticle. A particle-coagulation phenomenon occurred in the second stage of the nucleation reaction [26,27]. In the coagulative nucleation process, mature polymer particles aggregate between these later to produce a coalescent particle in a soap-free polymerized system [14,28,29,30,31]. Coagulation occurs as a phenomenon to stabilize the electrostatic charges in the system [32]. The main features of coagulated systems according to industrial scaling include the production of narrow and large-sized polymer particles and truly stable latex particles as byproducts [31,33,34,35]. The main advantage of monomer-starved semicontinuous polymerization is that the addition rate controls the polymerization rate. Due to this fact, the monomer addition rate directly affects the particle number. The particle nucleation effect was considered by some authors to be of secondary importance because the majority of copolymerized systems are analyzed at a low solid content in which the most soluble monomer governs the particle nucleation at the highest proportions [9,36]. In order to generate new industrial processes that are economically viable, solid content in copolymerization must rise. Thus, the nucleation process must be explained in terms of this novel focus.

A special remark must be included regarding coagulative nucleation during a semicontinuous emulsion polymerization in which—despite the use of a surfactant concentration above CMC—the system does not behave as monomer-starved but instead as monomer-flooded. The latter phenomenon was first approached by Sajjadi’s group [37], who mentioned that there were not sufficient micelles in the continuous phase to solubilize the monomer added. Typically, the monomer-flooded effect is presented in batch or semibatch polymerization, which is especially affected by the initial monomer charge identified by a secondary nucleation process and a bimodal particle size distribution [37]. The monomer-flooded model presented by Gilbert’s group [38] showed an improvement in the polymerization mechanism by means of a molecular weight reduction. A later model published by Sajjadi [39] revealed that a monomer-flooded system can stochastically move into a monomer-starved system until a dominant kinetic mechanism prevails due to polymer chains achieving a critical length. 

Industrial interest in modifying the rigidity and crystalline properties of PMMA through a union with elastomeric polymers (in which PBA is one of the most interesting options in the polymer industry) is remarkable. In this research, we report the particle size and morphology, polymerization kinetics, and nucleation mechanism in particle growth of poly(methyl methacrylate-*co*-butyl acrylate) (P(MMA-*co*-BA)) synthesized via SEHP with different water-solubility monomers under monomer-starved conditions. By controlling the monomer-feeding rate and the surfactant concentration, it was possible to correlate particle growth with the nucleation mechanism in order to visualize future industrial application.

## 2. Materials and Methods

### 2.1. Materials

MMA and BA monomers (≥99.0% pure) were obtained from Sigma (St. Louis, MO, USA). Methyl ester hydroquinone was removed from monomers with a DH-4 distillation column (Scientific Polymer Products, Ontario, NY, USA). Sodium dodecyl sulfate (SDS, 98.0% from JT Baker), potassium persulfate (KPS, 99.0% from Sigma, St. Louis, Missouri, United States), sodium bicarbonate (SBc, 99.7% from Jalmek, San Nicolás de los Garza, Nuevo León, México), bi-distilled and deionized water were also used. Comonomer addition to the reaction system was controlled by a syringe pump (kdScientific, Holliston, MA, USA).

### 2.2. Experimental Latex Preparation

Polymerizations were carried out via SEHP using different comonomer feeding rates and SDS contents. First, 130 g of water and SDS were added and mixed in a reactor of 500 mL (two concentrations of SDS were used in the two copolymerization processes (1 and 2 wt %) based on the total of 150 g of MMA and BA), and then 0.45 g of SBc was charged to a reactor and heated at 70 °C under a nitrogen atmosphere with continuous stirring (350 rpm) for 60 min. After 1 wt % of KPS (based on the total of 150 g of MMA and BA) was solubilized in 20 g of water, the KPS solution was added to the reactor and maintained for 30 min. Finally, the MMA and BA mixture was fed into the reactor at a controlled rate, which was different in each process of copolymerization (0.5, 1.0, and 2.0 g min^−1^). In all polymerizations, the weight ratio of MMA to BA was maintained constant (*weight*_MMA_/*weight*_BA_ = 1.5), and the total amount of the comonomer mixture was 150 g. Subsequently, all the reactions were maintained under the same conditions for 60 min after the comonomer addition was completed. Batch emulsion polymerization (BEP) was carried out under similar conditions to those of the procedure and at amounts of reagents described previously except that 150 g of MMA/BA was added to the reactor after the SDS addition. Subsequently, the initiator was added to allow the reaction to proceed for 3 h.

### 2.3. Analysis

Sampling aliquots (4 mL) were extracted at different intervals with a pipette according to polymerization conditions to measure the particle size and conversions gravimetrically. Dynamic light scattering (DLS) equipment was used to measure the particle size and Z-potential in a Malvern Zetasizer ZS90 apparatus (Malvern Panalytical Ltd., Malvern, UK) The latex samples were placed in a glass cuvette to measure the particle size and in disposable folded-capillary cells to measure the Z-potential; these samples were diluted with water up to 100 times to avoid multiple scattering and undiluted, respectively. The number of particles (*Np*) was calculated using Equation (1) [2]:(1)$$N_{p}=\frac{6R_{a}tx_{i}}{\pi \rho_{pol}D_{pol}{}^{3}V\left(t\right)}$$
where *R_a_* is the addition rate; *x_i_* is the instantaneous conversion; *D_pol_* is the diameter measured at different times in DLS; *V(t)* is the total volume in the reactor at a determined time; *ρ_pol_* is the average density of each polymer in the reactor ($\rho_{pol}=F_{1}\rho_{1}+F_{2}\rho_{2}$, where *F*_1_ and *F*_2_ are the wt % of polymer in the experiment); and *ρ*_1_ and *ρ*_2_ are the polymer densities of PMMA and PBA, respectively. The PMMA and PBA densities considered were 1.2 g mL^−1^ and 1.08 g mL^−1^, respectively.

The surface coverage ratio (*Sc*) of the surfactant on the particles (*r*) was the surface covered by the available surfactant molecules (*A_s_*) over the total surface area of polymer particles (*A_p_*) according to Equation (2) [5]:(2)$$r=\frac{A_{s}}{A_{p}}=\frac{c_{s}N_{Av}a_{s}}{\pi D_{p}^{2}N_{p}}$$
where *c_s_* is the surfactant concentration, *N_Av_* is Avogadro’s number, *a_s_* is the molecular surface area value for SDS (0.5 nm^2^ molecule^−1^) [40], *D_p_* is the particle size from the z-average diameter, and *N_p_* is the number of particles.

The particle coagulation rate data were fitted with the Smoluchowski equation for the particle coagulation kinetics of N_p_ as shown in Equation (3):(3)$$-r_{N}=-\frac{dN_{p}}{dt_{r}}=K_{c}N_{0}^{2}$$
where $-r_{N}$ is the particle coagulation rate, *K_c_* is the coagulation coefficient, *N_0_* is the particle number, and *t_r_* is the normalized time (*t*/*t_∞_*, where *t_∞_* is the total addition time).

### 2.4. Characterization

#### 2.4.1. Transmission Electron Microscopy

A JEOL 1010 transmission electron microscope (TEM, JEOL Ltd., Tokyo, Japan) was employed to observe the morphology of the copolymer particles. To accomplish this, one drop of latex was diluted up to 200 times with water, and one drop of this solution was deposited onto a copper grid and then dyed with a drop of 1.0 wt % phosphotungstic acid aqueous solution for 24 h and dried in a vacuum oven at 50 °C for 12 h.

#### 2.4.2. Glass Transition Temperature

The glass transition temperature (Tg) was measured via differential scanning calorimetry (DSC) in order to examine the Tg in the copolymer formed. A Perkin Elmer Pyris 6 (Norwalk, CT, USA) differential scanning calorimeter was used. A nitrogen purge gas flow of 50 mL min^−1^ and a heating rate of 10 °C min^−1^ were carried out in all runs. A second scan of each sample under the same conditions was selected for the calculation of glass transition temperatures to eliminate thermal history.

#### 2.4.3. Nuclear Magnetic Resonance

The NMR spectra of the copolymers were obtained using JEOL equipment (JNM-ECA600) spectrometer (JEOL Ltd., Tokyo, Japan) of high resolution in CDCl_3_ (^1^H: 600 MHz). The 1D NMR spectra of the copolymers were recorded at a frequency of 600 MHz via ^1^H and were employed to calculate the compositions and stereoregularity of the copolymers.

## 3. Results and Discussion

The SEHP was performed at feeding rates (*R_a_*) of 0.5, 0.1, and 0.2 g min^−1^ for 1.0 and 2.0 wt % SDS. Both obtained latexes (via SEHP and BEP) were cloudy, which is typical of direct emulsions; however, the latexes at 2.0 wt % SDS obtained via SEHP showed a lower particle diameter (107 nm) and were somewhat more bluish than those obtained by BEP (128 nm), which will be further discussed with regard to the DLS measurements.

### 3.1. Comonomers Conversion

Figure 1 shows the instantaneous conversions (*x_i_*) as a function of the relative time (*t_r_*) for 1 (Figure 1a) and 2 (Figure 1b) wt % SDS and the addition rates of 0.5, 1.0, and 2.0 g min^−1^. *t_r_* was defined as the ratio of addition time (*t*) divided by the total addition time (*t_∞_*) at each *R_a_* used in the polymerization. Equation (4) was used to calculate *x_i_* [2,6]:(4)$$x_{i}\left(t\right)=\frac{w_{pol}}{w_{mon}}$$
where *w*_pol_ is the weight of polymer at a time *t* and *w_mon_* is the weight of monomer added at a time *t*, while *x_i_* and the global conversion (*X*) (Figure 2) were calculated from gravimetric measurements with aliquots taken at different times. The copolymerizations were fast for all the feeding rates studied compared with the BEP process. In Figure 1a, it can be observed that at low *t_r_* (~0.085), a fast comonomer conversion took place with *x_i_*~0.7 that slightly grew between *t_r_* = 0.1 and 0.35. Afterward, there was a gradual increase until reaching final conversions higher than 0.9. For *R_a_* = 2 g min^−1^, the copolymerization was slightly slower and reached a comonomer conversion of *x_i_* = 0.8 at *t_r_*~0.2; in the BEP process, there was a delay in the conversion to a few minutes of polymerization.

At short reaction times, there were enough radicals and surfactant for particle nucleation; as the reaction progressed, the more soluble monomer (MMA) continued the nucleation in the aqueous phase to maintain the growth of the polymer chains. With both, it is possible that a great amount of BA diffused into the micelle, while the polymerization of BA inside the micelle could be delayed until PMMA-PBA co-oligomers entered into micelle, which caused the accumulation of BA and a decrease in the conversion in this time period. Mainly, when there was not enough surfactant to stabilize or nucleate the nanoparticles, this led to a decrease in the rate of polymerization and an accumulation of the monomer in the micelles (Figure 3). Then, we could assume that the SDS concentration of 1.0 wt % worked under flooded conditions while the lowest feeding rate exhibited the monomer-starvation conditions. In the homopolymerization process carried out with the monomers via SEHP for the addition rate of 2 g min^−1^ and 1.0 wt % of surfactant, the polymerization of MMA obtained a high conversion (*xi*) and *R_p_* (Appendix A), but at *t_r_*~0.4, the particles were agglomerated and collapsed due to surfactant deficiency where *Sc* was lower (Appendix A). Therefore, the dominant mechanism of polymerization for MMA was homogenous nucleation. However, in the BA polymerization under the same conditions from kinetics at low times, an amount of BA added was diffused inside the micelles because of decreased conversion and an increased residual monomer at a low *t_r_* (0.1). Then, an increase in particle size caused by swelling of the micelles and *Sc*~0.44 was sufficient to maintain colloidal stability throughout the reaction via diffusion of the monomer to the particles in formation (Appendix A). On the contrary, for 2.0 wt % of SDS content (Figure 1b), full starvation conditions were achieved because *x_i_*~0.9 at *t_r_*~0.8 for *R_a_* = 2 g min^−1^. This was due to the presence of a greater amount of surfactant that allowed stabilizing of the monomer droplets and the formation of new particles. This effect gave rise to an additional pathway of comonomer diffusion, especially at short reaction times. At 1.0 wt % of SDS, *x_i_* showed a delay in the conversion, suggesting that the low concentration was insufficient to solubilize the comonomers (mainly BA) due to the low solubility in water, which diffused into the micelles at a very early stage of copolymerization. This was corroborated by the concentration of 2.0 wt % of SDS, in which this delayed induction time was not observed.

Figure 2 shows the global conversions (*X*) as a function of *t_r_* for 1.0 (Figure 2a) and 2.0 wt % SDS (Figure 2b), and the addition rates of 0.5, 1.0, and 2.0 g min^−1^. *X* were obtained as in Equation (5) [41]:(5)$$X\left(t\right)=\frac{w_{pol}}{M_{total\;}}$$
where *M_total_* is the weight of total monomer added.

A continuous and progressive polymerization was observed until the final amounts of comonomers were added. For BEP, *X* achieved its maximal conversion (*X*~0.92) at *t_r_*~0.4, and then remained constant. SEHP copolymerizations displayed smaller error bars, in comparison with BEP copolymerizations; therefore, there was more control of the reaction by SEHP.

The relative time evolution of the residual monomer weight fraction of SEHP at different *R_a_* and SDS concentrations is shown in Figure 3. Figure 3a shows a tendency to accumulate the monomer for an *R_a_* of 0.5 g min^−1^ and 2 g min^−1^, which demonstrated that the system could not polymerize the full amount of monomer added. At the lowest *R_a_*, there was migration selectivity due to the water solubility of monomers (MMA is almost 14 times more soluble than BA) [42]. The solubility of MMA and BA in water was 150 mM and 11 mM (from 25 to 50 °C), respectively [42]. Thus, MMA could be maintained at a higher number of nucleated particles in the aqueous phase, while BA migrated into the micelles to later nucleate particles via micellar nucleation. For the highest feeding rate and the lowest surfactant concentration, the highest accumulation of monomers was achieved; this behavior was similar to the polymerization reaction under flood conditions. However, Figure 3b shows that the starvation conditions were improved in all of the SEHPs due to the increased amount of surfactant available to stabilize the large number of copolymer particles formed in the system.

### 3.2. Polymerization Rate Analysis

The comparison between the ratio of the polymerization rate (*R_p_*) and *R_a_* (*R_p_*/*R_a_*) of the two concentrations of surfactant and their comonomer feeding rates as a function of *t_r_* is depicted in Figure 4. The polymerization rate (*R_p_*) or comonomer conversions in the polymer were dependent on the comonomer addition rate (*R_a_*) [43,44]. The instantaneous polymerization parameters measured whether monomer-starved conditions were achieved. According to Wessling et al. [43], *R_p_* _≈_ *ϕ_p_ R_a_* (where *ϕ_p_* is the volume fraction of polymer in the monomer-swollen particles). Under monomer-starved conditions, *R_p_* ≈ *R_a_*_,_ as has been published elsewhere [2,6]. In Figure 4a, a behavior *plateau* was observed for *R_p_*/*R_a_* < 0.9 between *t_r_* values of 0.1–0.35 for 1.0 wt % of SDS and for all feeding rates. As the polymerization progressed, *R_p_*/*R_a_* increased continuously until reaching values higher than 0.9. In this period, nucleation could occur under monomer-flooded conditions via monomer accumulation in short polymerization times [6], which was mainly due to the low solubility of BA in water, which limited its nucleation in the aqueous phase. On the other hand, its counterpart with the highest surfactant concentration did not present in this region (Figure 4b). This phenomenon of localization during the nucleation stage could be explained by a stabilization of the particles. The low amount of surfactant available to stabilize the larger number of particles led to (for a certain time period) *R_p_* not being as fast as the comonomer addition. Studies on the homopolymerization of MMA and BA in emulsion showed that at the beginning of the reaction in the aqueous phase, both monomers had very different kinetics at low concentrations of surfactant. Sajjadi and Brooks [45] performed the polymerization of BA by SEHP. These authors showed long periods of inhibition and low *R_p_*/*R_a_* only when there was a high ratio of emulsifier added throughout the reaction. This reaffirmed that poorly soluble monomers in an aqueous phase cannot nucleate easily and that the higher amount of monomer diffuses into micelles until they are nucleated by growing oligomeric radicals in the aqueous phase. Otherwise, in the kinetics of the emulsion polymerization of MMA under similar reaction conditions, no inhibition periods would have been observed, and polymerization would have been carried out under flooded conditions [46]. Thus, the just-formed copolymer particles were forced to join with mature particles in order to preserve latex stability. This did not imply changes in the reaction sites or differences in the conformation of the final polymer chain structure.

### 3.3. Particle Size and Surfactant Coverage Ratio

Figure 5 shows the *z*-average particle size (*Dp_z_*) of P(MMA-co-BA) nanoparticles (NPs) synthesized via SEHP as a function of *X* for the three feeding rates studied and in the presence of 1.0 and 2 wt % of SDS as measured by DLS at 25 °C. Experiments carried out with the comonomer addition at 1 and 2 g min^−1^ presented similar behaviors: the particle size increased as *X* increased (Figure 5a). All the systems contained 50 wt % of solid content, which allowed identifying the solid content point at which a desired particle size was obtained. At the end of the reaction, values of *Dp_z_* were very close to those obtained via BEP due to the coagulation of copolymer particles. On the other hand, for 2.0 wt% of SDS (Figure 5b), significant differences were observed between the feeding rates within the *X* range of 0.5–0.9. The final *Dp_z_* for the NPs produced via SEHP were smaller (~20 nm) for all addition rates studied than those obtained via BEP. The increase in *Dp_z_* with the increase in conversion was highly related to particle coagulation. However, for NPs obtained via BEP, *Dp_z_* was practically constant from the initial reaction stage until final conversion because micellar nucleation was predominant. When the emulsion homopolymerization was carried out by means of monomer-starved conditions, the *Dp_z_* was maintained constant because nucleation in the aqueous phase increased the number of particles in the latex [6]. However, in the NPs obtained via SEHP, a continuous increase was observed in *Dp_z_*. Hence, we proposed that the micellar nucleation mechanism was induced by BA due to its low solubility, while MMA was induced by homogeneous nucleation. This allowed us to infer that the mechanism that controls the polymerization process is coagulative nucleation because NPs formed in the aqueous phase were incorporated onto the surface of NPs formed by micellar nucleation, as was further clarified by TEM images. This phenomenon is commonly observed in one-step emulsion polymerization [27].

Figure 6 shows the *Sc* as a function of *t_r_* for the copolymerization of MMA and BA with different feeding rates. *Sc* shows the grade to which surfactant molecules stabilized the particles in the polymerization reaction. BEP is generally known to possess more surface coverage with surfactant because all polymerization is controlled by the mechanism of micellar nucleation [47]. Lower values of *Sc* were found for feeding rates of 0.5 and 1.0 g min^−1^ (Figure 6a). Nevertheless, for the addition rate of 2.0 g min^−1^, different behavior was observed: *Sc* was higher compared to the remaining two, and it increased at the end of *t_r_*. As can be observed, a higher *Dp_z_* and a lower number of NPs indicated that a greater amount of monomer was diffused into the micelle, and chain growth occurred to leave a more surfactant-free surface. As shown in Figure 6b, *Sc* values of BEP remained high (~0.5). For SEHP, the *Sc* values were lower compared to BEP due to the large number of particles formed. *Sc* showed a minimum at *t_r_* of 0.6 for all rates at 1.0 wt % of SDS followed by an increase in *Sc*; whereas for the concentration of 2.0 wt % SDS, a slight increase was observed after the minimum. Therefore, the increase in *Sc* after 50% of the conversion indicated that the nucleation mechanism was controlled for both micellar and homogeneous nucleation to stabilize the particles in the system [47]. However, in the BEP reaction, after *Sc* reached the minimal value, this remained constant for both surfactant concentrations, indicating that micellar nucleation was controlled throughout the entire polymerization process.

To explain the coagulation phenomenon in this study, we measured the zeta potential (Z-potential) at different times in the BEP reaction and at 2.0 g min^−1^ of comonomer feed for 1.0 wt % of SDS in SEHP (Appendix A). The Z-potential showed an increase from −69 to −47 mV throughout the BEP reaction. This increase can be explained by the large particle sizes and the slight increases in the particle number, which was caused by comonomer droplets in uninitiated micelles moving toward the growing nucleated micelles from the beginning to the end of the polymerization. Meanwhile, for SEHP, the Z-potential remained at low values during the initial times (between −66 mV and −70 mV) caused by the nucleation of new particles (mainly in the aqueous phase) and then began to increase from −62 mV to a *t_r_* of 0.5 due to the increase in the particle size caused by the presence of monomers inside of the micelle that began to polymerize. After this time, this increase became pronounced and reached its maximum at −56 mV to a *t_r_* of 0.7; in this time range, we propose that secondary particles began to coalesce on the primary particles. Subsequently, the Z-potential decreased again, and new nucleated particles reattached to growing particles. This increase was in agreement with the results depicted in Figure 5 and Figure 6 for *Dp_z_* and for the minimal value in *Sc*, respectively, as well as for the particle number (*N_p_*) as shown in Figure 6. However, this process is very common in the use of charged particles [48].

### 3.4. Coagulation Mechanism Analysis

The mechanism of particle formation indicates how the process of obtaining primary small particles takes place throughout the swelling and growth in the polymerized system. Figure 7 presents the behavior of the number of particles (*N_p_*) during the reactions for all polymerization systems. For the SEHP, the tendency indicated that at low *X* values, high values of *N_p_* were achieved. Nonetheless, from propagation until the termination reaction was processed, *N_p_* diminished by one order of magnitude. This decrease in particle number may have been caused due to the coagulative nucleation that occurred under monomer-starved or flooded conditions; therefore, it was possible to obtain up to 50.0 wt % solid content. Under monomer-starved conditions, when the homogeneous nucleation process was dominant, *N_p_* increased constantly because the size could be controlled. Moreover, in emulsion polymerization, when there is particle coagulation, *N_p_* is controlled by two steps: a fast step in which particle aggregation occurs and a slow step in which particle coalescence controls the kinetics [27]. In our research, these steps were observed by analysing the *N_p_*; however, the coalescence process occurred in which *N_p_* decreased slightly as *X* increased because comonomers were added throughout the entire process of copolymerization.

The insets in Figure 7a,b show the particle coagulation rate $\left(dN_{p}/dt\;\right)$ versus *X* for the *N_p_* carried out via SEHP at different addition rates and with 1.0 and 2.0 wt % of SDS content and BEP. As can be observed, as *X* increased, $dN_{p}/dt$ decreased rapidly until reached a minimum (indicated with an arrow) at *X*~0.11 and *X*~0.17 for SEHP with 1.0 and 2.0 wt % of SDS content, respectively. The decrease during this period was highly related to the continuous formation of particles or to the predomination of homogeneous nucleation in that there were many radicals to form oligomeric or polymeric radicals in an aqueous phase to nucleate particles. Subsequently, there was a slight increase in $dN_{p}/dt$ until *X*~0.5 caused by a diffusion of comonomers toward the particles present or those nucleated. At higher values of *X (>~0.5)*, a plateau was reached. These results infer that the mechanism that controlled the kinetics was coagulative nucleation and that it depended mainly on the monomer- and surfactant-starved conditions in the systems. As discussed previously, by not possessing enough surfactant molecules to stabilize the monomer droplets entering the system, the newly formed particles coalesced to maintain the colloidal stability of the latex. In addition, the reaction achieved stability in the particle-formation process thanks to the redistribution of surfactant molecules from the previously formed micelles to newly formed particles.

### 3.5. Thermic Copolymer Behavior

The glass transition temperatures (T_g_) of copolymers synthesized via BEP and SEHP at different concentrations of SDS and monomer feed rates were obtained by using a DSC analysis as shown in Figure 8. In the thermograms, it can be appreciated that the T_g_ of the copolymer in BEP for two SDS concentrations was obtained at ~20 °C and that for the SEHP it was between 21 and 23 °C (Appendix A). This difference in the T_g_ can be explained by the manner in which the polymerization was carried out in both processes. In BEP, in the first step, the diffusion process of monomers into micelles to be swollen took place; notwithstanding this, the ratio of MMA/BA monomers and oligomeric radicals in the aqueous phase increased during the initiation step because the solubility of MMA in water was higher than in BA. When the copolymeric radicals moved into the micelle to carry out the micellar nucleation, the relationship was maintained; this allowed synthesizing copolymer chains with higher sequences of MMA chemical bonds and a consequent increase in the tacticity.

### 3.6. Stereoregularity Analysis

Using ^1^H-NMR spectroscopy, it was possible to identify the formation, stereoregularity of *MMA*, and the monomer composition that was present in the copolymers of some polymerization processes of SEHP and BEP (Figure 9). The tacticity of *MMA* was determined by integrating intensity peaks of α-methyl protons at 0.80, 0.99, and 1.10 ppm (Appendix A), which corresponded to the syndiotacticity (*rr*), heterotacticity (*mr*), and isotacticity (*mm*), respectively. The copolymers with a higher content of *mm* were those obtained via SEHP (~8%), while BEP was 3.1%. These higher values of *mm* were related to the increase in Tg of the copolymer via SEHP and were higher than those obtained via BEP. In this process, the polymer Tg was dependent on both monomers; however, it was also related to the stereoregularity of the polymer [2]. Monomer compositions in the copolymer were calculated using the area of signal peaks of the methylene proton of PBA and methyl proton PMMA (Equation (6)), thereby obtaining a composition of *BA* for the SEHP process of 21% and for BEP of 20%.
(6)$$C_{BA}=\frac{\frac{A_{BA}}{2}}{\frac{A_{BA}}{2}+\frac{A_{MMA}}{3}}\times 100$$

### 3.7. Copolymer Particles Morphology

The TEM images of copolymer particles shown in Figure 10 were taken at 120 min of reaction time (Figure 10a,c) and at the end of the copolymerization (300 min) (Figure 10b,d) at the addition rate of 0.5 g min^−1^ for SEHP at 1 and 2 wt %. In this figure, rows 1 and 2 show the NPs formed at 1.0 and 2.0 wt % of SDS content, respectively. In Figure 10a,c, the beginning of the coagulation step can be observed in which smaller spherical particles began to adhere to larger particles. Moreover, at a low concentration of SDS, the particle size as shown by TEM (*Dz*_0_) at 120 min and 300 min were 70 and 150 nm, respectively; whereas at the higher concentration, *Dz_0_* at 120 and 300 min were 55 and 110 nm, respectively. Note that the *Dz_0_* of particles produced with 1.0 wt % of SDS (Figure 10a,b) were larger than those obtained with 2.0 wt % of SDS (Figure 10c,d). The increase in *Dz_0_* in a coagulation process due to the collision frequency of particles and the subsequent merging process is commonly denominated as flocculation [49]. Contrariwise, the image of the particles produced via BEP shows a higher *Dz_0_* and high polydispersity (Appendix A).

## 4. Conclusions

High solid content (50%) with stable latexes via SEHP under monomer-starved conditions was obtained. Under these conditions, the mechanism of coagulative nucleation was demonstrated by *Dp_z_, N_p_*, *Sc*, Z, and the Smoluchowski model. Moreover, almost all reaction systems studied were achieved under these conditions except at the concentration of 1 wt % at 2 g min^−1^, in which flooded conditions dominated and the radicals could not consume the greater amount of comonomers that were added because there was insufficient surfactant for nucleating particles. In addition, the low solubility of BA contributed to a faster diffusion of BA inside the micelle, contributing to the decreases in Rp and an increase in the monomer accumulation. The particle size was maintained as a linear increase as a result of the continuous swelling and nucleation of the particles; this was corroborated by a decrease in Np and Sc until 50% of conversion. After this, a more pronounced increase in the particle size was observed at higher conversions because the new particles that formed coalesced on primary particles. In this range, we observed a slight increase in the Sc, thereby requiring less surfactant to stabilize the particles, which was corroborated by the TEM images. The Z-potential exhibited an increase after 40% of global conversion until reaching a maximal value of −56 mV. Therefore, particles nucleated via homogenous nucleation were agglomerated or coalesced on mature particles, and the Z-potential decreased again as a result of the homogeneous nucleation of particles due to the continuous addition of comonomers until the final polymerization process. Based on the particle coagulation of the rate, it was possible to identify the coagulative nucleation to global conversions higher than 40% because there was a minimal change in the particle number caused by continuous aggregation of the new particles to the mature particles. This behavior also was demonstrated by the Smoluchowski equation, which also demonstrated a linear behavior at higher conversions. Therefore, three regions were observed in the SEHP process, two predominated by nucleation mechanism: the homogeneous nucleation, transition zone, and coagulative nucleation. In addition, in the final polymerization process, certain results were similar (such as conversion and particle size), while the Tg, stereoregularity of the copolymer, and stability of the latex were higher in the SEHP process due to the addition rate and coagulative nucleation, which are very important in the industrial process because it is possible to maintain the stability of latex for a longer time. Finally, the SEHP copolymerization process allowed obtaining a high solid content with high colloidal stability and a Tg close to room temperature. These characteristics enhance the use in the formation of films (mainly for paints and adhesives). The results of this study can be a tool to modify variables according to the required final product.

## Figures and Tables

**Figure 1 polymers-15-01628-f001:**
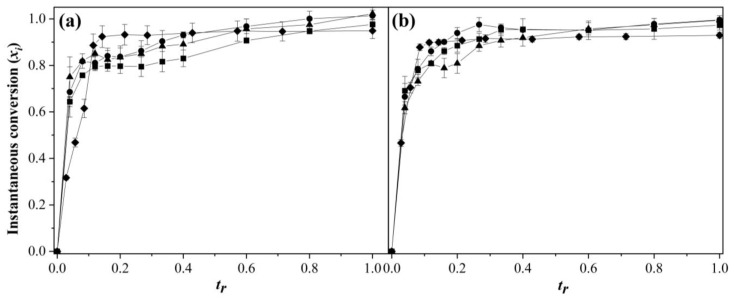
Evolution of instantaneous conversion as a function *t_r_* in the copolymerization of MMA-BA prepared via SEHP with (**a**) 1.0 and (**b**) 2.0 wt % of SDS content at different feed rates according to (■) 2.0 g min^−1^, (●) 1.0 g min^−1^, (▲) 0.5 g min^−1^, and (♦) BEP.

**Figure 2 polymers-15-01628-f002:**
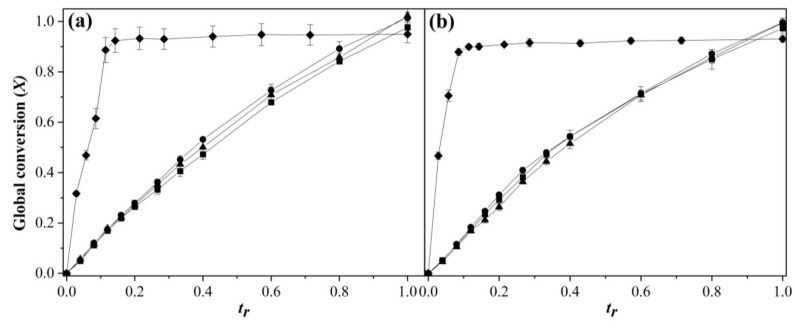
Evolution of global conversion as a function of *t_r_* in the copolymerization of MMA-BA prepared via SEHP with (**a**) 1.0 and (**b**) 2.0 wt % of SDS content at different feed rates according to (■) 2.0 g min^−1^, (●) 1.0 g min^−1^, (▲) 0.5 g min^−1^, and (♦) BEP.

**Figure 3 polymers-15-01628-f003:**
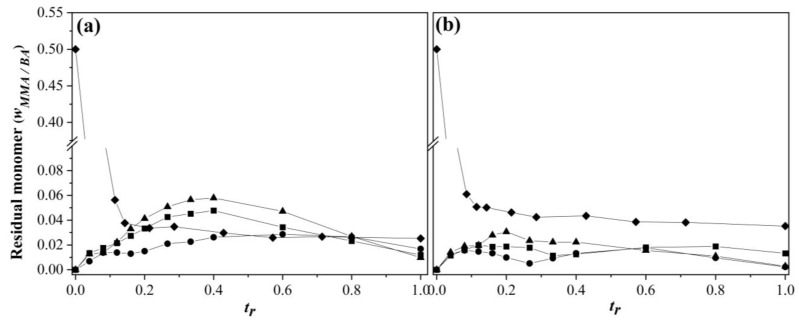
Evolution of the residual monomer weight fraction as a function of *t_r_* in the copolymerization of MMA-BA prepared via SEHP with (**a**) 1.0 and (**b**) 2.0 wt % of SDS content at different feed rates according to (■) 2.0 g min^−1^, (●) 1.0 g min^−1^, (▲) 0.5 g min^−1^, and (♦) BEP. Note that in (**a**), BEP is depicted by (◊) in order to improve identification.

**Figure 4 polymers-15-01628-f004:**
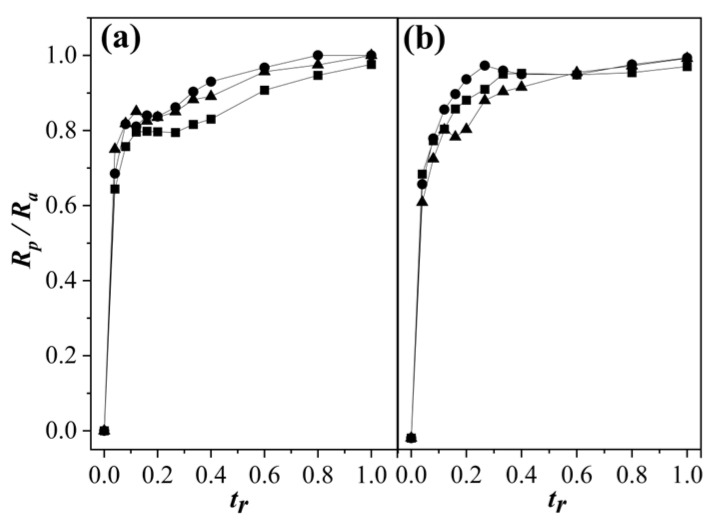
Ratio of the instantaneous polymerization and monomer addition rates (*R_p_*/*R_a_*) vs. *t_r_* in the copolymerization of MMA-BA with (**a**) 1.0 and (**b**) 2.0 wt % of SDS content at different feed rates according to (■) 2.0 g min^−1^, (●) 1.0 g min^−1^, and (▲) 0.5 g min^−1^.

**Figure 5 polymers-15-01628-f005:**
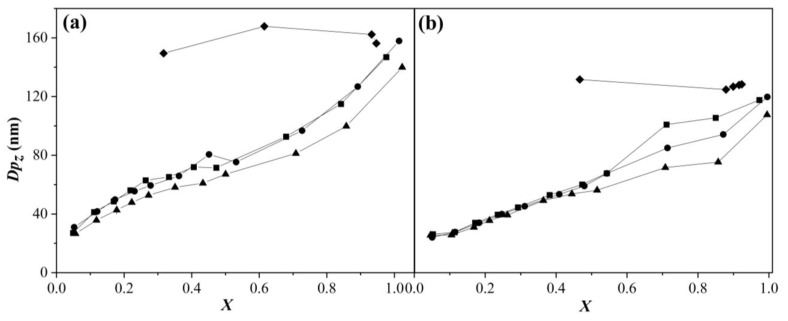
*z*-Average particle size versus global conversion in the copolymerization of MMA-BA prepared via SEHP with (**a**) 1.0 and (**b**) 2.0 wt % of SDS content at different feed rates according to (■) 2.0 g min^−1^, (●) 1.0 g min^−1^, (▲) 0.5 g min^−1^, and (♦) BEP.

**Figure 6 polymers-15-01628-f006:**
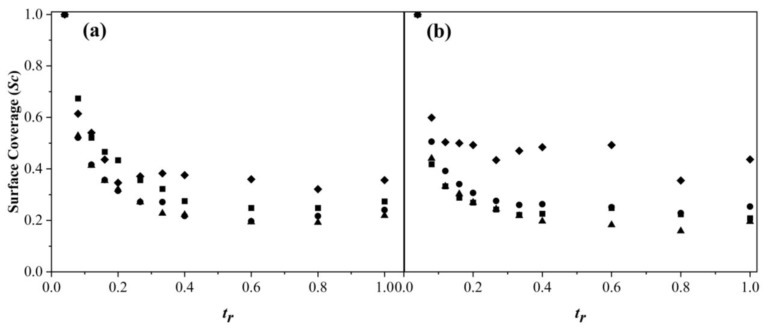
Surface coverage as a function of *t_r_* in the copolymerization of MMA-BA prepared via SEHP with (**a**) 1.0 and (**b**) 2.0 wt % of SDS content at different feed rates according to (■) 2.0 g min^−1^, (●) 1.0 g min^−1^, (▲) 0.5 g min^−1^, and (♦) BEP.

**Figure 7 polymers-15-01628-f007:**
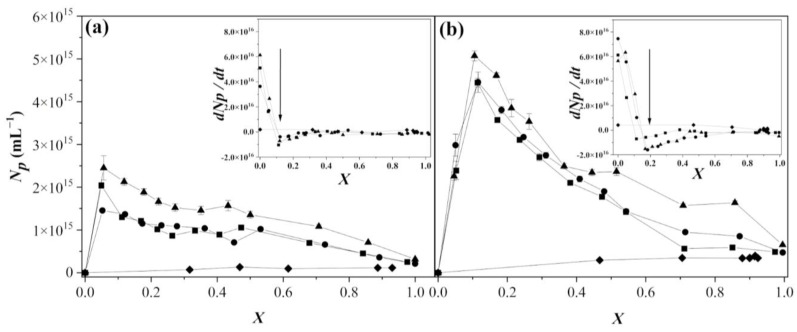
Number density of particles as a function of global conversion in the copolymerization of MMA-BA prepared via SEHP with (**a**) 1.0 wt % and (**b**) 2.0 wt % of SDS content at different feed rates according to (■) 2.0 g min^−1^, (●) 1.0 g min^−1^, (▲) 0.5 g min^−1^, and (♦) BEP. Inset: $r_{N}$ as a function of global conversion.

**Figure 8 polymers-15-01628-f008:**
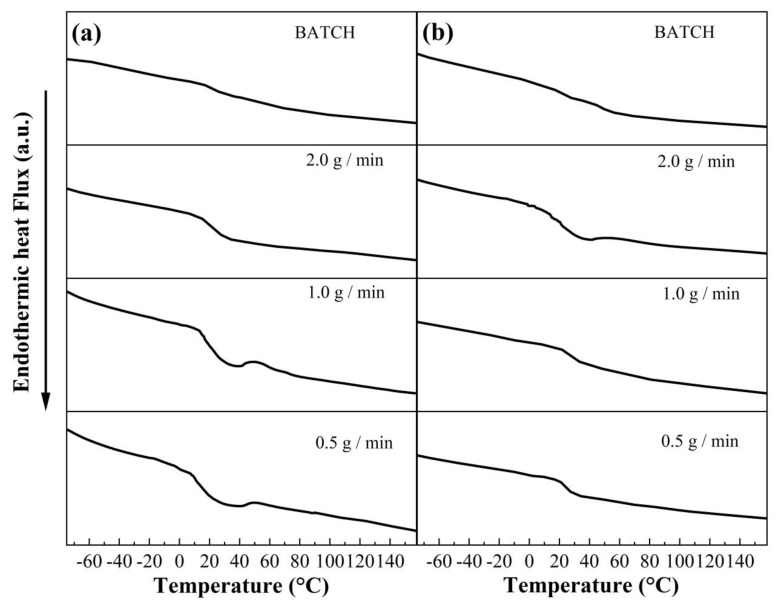
DSC thermograms of copolymers of MMA-BA prepared via batch emulsion and SEHP at (**a**) 1.0 wt % and (**b**) 2.0 wt % of SDS and different addition rates of 0.5, 1, and 2 g/min.

**Figure 9 polymers-15-01628-f009:**
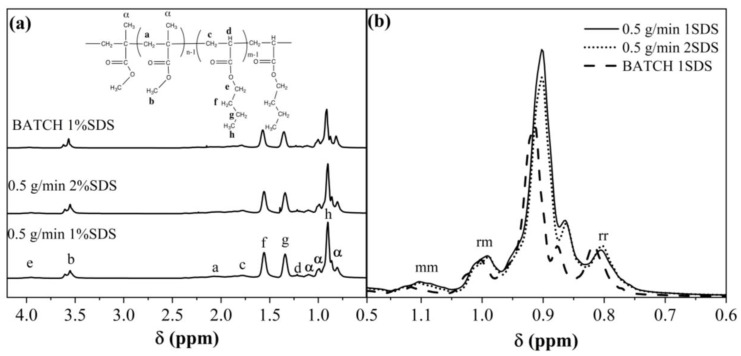
^1^H-NMR spectroscopy of selected samples to identify *rr*, *mr*, and *mm*.

**Figure 10 polymers-15-01628-f010:**
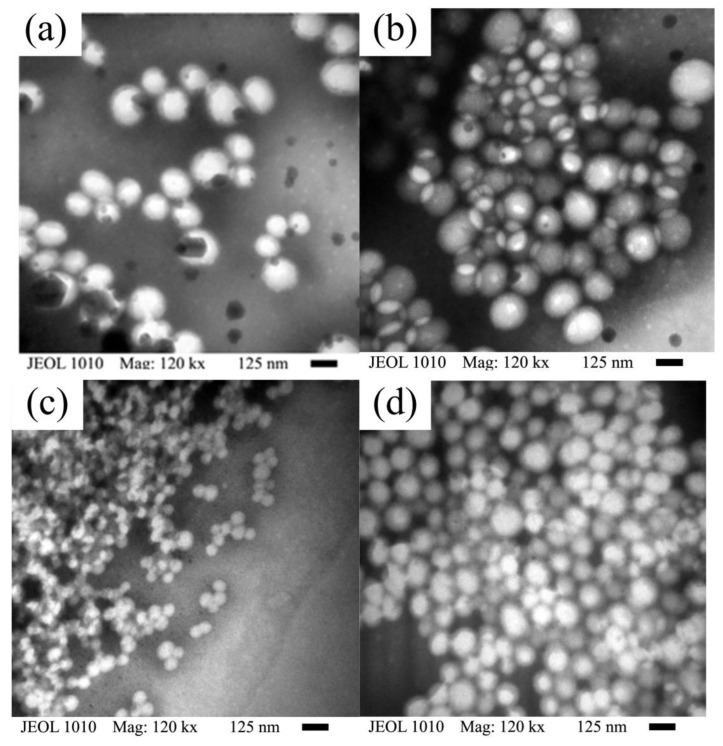
TEM images of copolymer particles synthesized via SEHP at *R_a_* of 0.5 g min^−1^ at an intermediate time of 120 min (**a**,**c**) and the final copolymer (**b**,**d**), with 1.0 (**a**,**b**) and 2.0 (**c**,**d**) wt % of SDS content, respectively.

## Data Availability

The data generated from this research are available from the authors.

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
