# Peer review of "Coagulative Nucleation in the Copolymerization of Methyl Methacrylate–Butyl Acrylate under Monomer-Starved Conditions"

_polymers, 2023, doi:10.3390/polym15071628_

Round 1

Reviewer 1 Report

The study by Castellanos et al. presents mechanism of coagulative nucleation after 50% feeds in the emulsion polymerization of MMA-BA under monomer starved conditions. The mechanism is observed by means of zeta potential, particle surfactant coverage area, particle coagulation rate, particle number and final particle size. Overall, this contribution will be of high interest to engineers and scientists in industry and academia. This manuscript can be accepted after the following questions are addressed. Below are some comments -

 COMMENTS

1.     What does Ra denotes in line 28 of the Abstract?

2.     Regarding the discussion on Page 10, what is the negative value of zeta potential after which the latex particles will undergo agglomeration or instability ?

3.     For the non-experts, can the authors clearly define difference between homogenous nucleation in first 40% feeds and the coagulative/heterogenous nucleation later ?

4.     Based on the manuscript, it appears that the monomer travels through the aqueous phase into the growing polymer particle, and then polymerizes. For these two steps, which one is the rate limiting step? Can the authors define rate constants for monomer travel in aqueous phase for BA and MMA?

5.     Several minor errors exist in the manuscript. These should be corrected.
Line 88, it should be “molecular weight”

Author Response

Coagulative nucleation in the copolymerization of methyl methacrylate-butyl acrylate by monomer-starved conditions

Sujey G. Castellanos, V. Vladimir A. Fernández-Escamilla, Miguel Á. Corona-Rivera, Karla J. González-Iñiguez, Arturo Barrera, Francisco J. Moscoso-Sánchez, Edgar B. Figueroa-Ochoa, Israel Ceja, Martín Rabelero and Jacobo Aguilar

Response to Reviewer # 1

We thank the reviewer for having examined the manuscript very carefully and for the remarks and suggestions for improving it.  We wish to resubmit a modified version of the manuscript, which includes the following suggestions.  We have made changes accordingly and given hopefully convincing replies. Also, the manuscript was carefully proofread for misspelling and typing errors.

Response to comments:

  1. What does Ra denotes in line 28 of the Abstract?

Re: The Ra was already described in the abstract: High instantaneous conversions (> 90%), and high ratio polymerization rate/addition rate (Rp/Ra) ≥ 0.9 are obtained at low times until the final copolymerization,

  1. Regarding the discussion on Page 10, what is the negative value of zeta potential after which the latex particles will undergo agglomeration or instability?

Re: There is no precise negative value of zeta potential when the particle begins to coalesce, however, at tr=0.5 correspond to the value of -62 mV, after a pronounced increase in the zeta potential, due to the decrease in the repulsion by coalescence of the particles. This behavior was corroborated by a decrease in the particles number (Np), an increase in particle size (Dpz), and an increase in surface coverage (Sc). Therefore, the discussion on zeta potential on page 10 was modified and its interpretation has been expanded.

  1. For the non-experts, can the authors clearly define difference between homogenous nucleation in first 40% feeds and the coagulative/heterogenous nucleation later?

Re: When monomer content in a system is low (<25.0 wt.%) it favors the homogenous distribution of molecules in the continuous medium (in this case aqueous phase) and allows the homogenous nucleation to occur. The polymerization dynamics change when the feeds are higher than 40%, the already formed particles are stabilized by coalescence and coagulative nucleation proceeds.  In several parts of the document such as lines 65 and 74, homogeneous nucleation is defined and details of the process are given. In the results section, line 238 from the comonomers conversion section and line 334 of the particle size section explained that the increase in particle size is highly related to particle coagulation. Additionally, the increment in surface coverage area related to particle coagulation was analyzed in line 364.

  1. Based on the manuscript, it appears that the monomer travels through the aqueous phase into the growing polymer particle, and then polymerizes. For these two steps, which one is the rate limiting step? Can the authors define rate constants for monomer travel in aqueous phase for BA and MMA?

Re: As Shahriar Sajjadi mentions in the research published in the Journal of Colloid and Interface Science 445 (2015) 174–182, kinetics analysis in polymerization is complicated when two monomers are present in the system. The determination of the polymerization rate of individual monomers in independent polymerizations is presented in Figure S1 as the polymerization rate analysis (polymerization rate/ addition rate). It was not the objective of this research to design an experiment to evaluate the rate constant of individual monomers in a combined polymerization. From Figure S1 it can be inferred that MMA mainly contributes to particle coagulation, limiting copolymerization reaction, and the presence of BA in micelles and particles contributes to colloidal stability in the final latex.

  1. Several minor errors exist in the manuscript. These should be corrected.
    Line 88, it should be “molecular weight”
    Re: Several minor errors were corrected in the manuscript

Reviewer 2 Report

Comments

1. The article is devoted to the emulsion copolymerization of methyl methacrylate with butyl acrylate. In the introduction, it should be indicated why this particular pair of monomers was chosen as the model system for the study?

2. In the introduction, it is necessary to explain what practical purpose this study pursues?

3. Key words should be supplemented with the main objects of research: methyl methacrylate and butyl acrylate.

4. The article discusses the different solubility of methyl methacrylate and butyl acrylate in water. Specify the specific values of the quantities at the synthesis temperature. To what extent does solubility affect the course of the process?

5. Copolymerization constants of methyl methacrylate and butyl acrylate differ significantly. How does this affect the parameters obtained in the work? Why was the ratio of monomers chosen exactly MMA/BA=1.5 in the studies?

6. The conclusions should include practical recommendations based on the research.

7. Notes on design:

Ø  On all figures it is necessary to remove several zero values at the origin. At the origin, there must be one common value for the axes, for example 0, or no values.

Ø  In the bibliography of this article, more than half of the articles are dated earlier than 2017. The list of references needs to be updated.

Author Response

Coagulative nucleation in the copolymerization of methyl methacrylate-butyl acrylate by monomer-starved conditions

Sujey G. Castellanos, V. Vladimir A. Fernández-Escamilla, Miguel Á. Corona-Rivera, Karla J. González-Iñiguez, Arturo Barrera, Francisco J. Moscoso-Sánchez, Edgar B. Figueroa-Ochoa, Israel Ceja, Martín Rabelero and Jacobo Aguilar

Response to Reviewer # 2

We thank the reviewer for having examined the manuscript very carefully and for the remarks and suggestions for improving it.  We wish to resubmit a modified version of the manuscript, which includes the following suggestions.  We have made changes accordingly and given hopefully convincing replies.

Response to comments:

  1. The article is devoted to the emulsion copolymerization of methyl methacrylate with butyl acrylate. In the introduction, it should be indicated why this particular pair of monomers was chosen as the model system for the study?

Re: From line 75 to 100, it is extensively indicated the importance to analyzed the polymerization mechanism of different water-soluble monomers, as MMA and BA are. This specific couple was chosen by the industrial interest of modified the rigidity and crystalline properties of PMMA through the union with elastomeric polymers, PBA is one of the most interesting options in polymer industry.

  1. In the introduction, it is necessary to explain what practical purpose this study pursues?

 Re: The comment was clarified in line 141.

  1. Key words should be supplemented with the main objects of research: methyl methacrylate and butyl acrylate.

Re: The words methyl methacrylate and butyl acrylate were added to the keywords.

  1. The article discusses the different solubility of methyl methacrylate and butyl acrylate in water. Specify the specific values of the quantities at the synthesis temperature. To what extent does solubility affect the course of the process?

Re: Data was found in range 25°-50°C. Marked on the line 291

To what extent does solubility affect the course of the process?

Re: Affect mainly the conversion and rate polymerization, because the monomers can diffuse inside the micelle, and will polymerize until monomeric or oligomeric radicals enter the micelle, decreasing conversion and rate polymerization, and consequence the accumulate monomer. 

This question appears explained in the pages 5 and 6, in the section 3.1. comonomers conversion.

  1. Copolymerization constants of methyl methacrylate and butyl acrylate differ significantly. How does this affect the parameters obtained in the work? Why was the ratio of monomers chosen exactly MMA/BA=1.5 in the studies?

How does this affect the parameters obtained in the work?

Re In fact, the constants of polymerization of both monomers differ significantly. However, to analyze how those affect parameters, a study deeper is necessary, and determine the kinetics constants in each step of the copolymerization process, which not were contemplated in this work

Why was the ratio of monomers chosen exactly MMA/BA=1.5 in the studies?

In this research, we wanted to get deeper into how the thermal parameters, such as transition temperature and stereoregularity were modified by the technique (SEHP). Mainly, to obtain a copolymer whose glass transition temperature was too close to the room temperature to form films, and increase the industrial application. Therefore, the comonomers proportion was fixed to 60/40. 

  1. The conclusions should include practical recommendations based on the research.

Re: The practical recommendations were redacted in the conclusions.

  1. Notes on design:

Ø  On all figures it is necessary to remove several zero values at the origin. At the origin, there must be one common value for the axes, for example 0, or no values.

Re: The figures were corrected and includes in the document

Ø  In the bibliography of this article, more than half of the articles are dated earlier than 2017. The list of references needs to be updated.

Re: References placed in this research are the basis of this work; there are no recent research works that can fundamentally contribute to our technique, comonomers system, or molecular design of our copolymer. Therefore, we can say that technique is novel because there is not work research recent on the technique SEHP, where there is a mechanism de coagulative nucleation. 

Reviewer 3 Report

The article "Coagulative nucleation in the copolymerization of methyl methacrylate-butyl acrylate by monomer-starved conditions" has important and new findings, but I consider that it requires a major review that highlights the importance of this technique, the results, be clear and Online with the correct visualization and analysis of the results of each study in an orderly manner, clarifying the importance of the information obtained. The document contains many minor errors and as presented it is difficult to understand and it is not clear with the results presented, this document does not have clear sections of the studies carried out in the characterization etc. Consider carefully enhancing your document.

-consider changing the word "photographs" to strictly "images" these are not photographs

-“Smoluchowsky” in “Smoluchowski”

-review the document especially the introduction, this should be more objective and brief, include the importance of this technique and its possible application

-check the units used in the methodology section

-please consider improving the wording of the methodology, it is incomplete and unclear, put the amount of monomers used followed by the rate as an indication

-divide the characterization into sections

-Improve figures and figure legends, they look incomplete or have errors

-include H-NMR spectra in the main text

Author Response

Coagulative nucleation in the copolymerization of methyl methacrylate-butyl acrylate by monomer-starved conditions

Sujey G. Castellanos, V. Vladimir A. Fernández-Escamilla, Miguel Á. Corona-Rivera, Karla J. González-Iñiguez, Arturo Barrera, Francisco J. Moscoso-Sánchez, Edgar B. Figueroa-Ochoa, Israel Ceja, Martín Rabelero and Jacobo Aguilar

Response to Reviewer # 3

We thank the reviewer for having examined the manuscript very carefully and for the remarks and suggestions for improving it.  We wish to resubmit a modified version of the manuscript, which includes the following suggestions.  We have made changes accordingly and given hopefully convincing replies. Also, the manuscript was carefully proofread for misspelling and typing errors.

Response to comments:

-consider changing the word "photographs" to strictly "images" these are not photographs

Re: The word collocated as photographs was changed by the word images, as requested

-“Smoluchowsky” in “Smoluchowski”

Re: The name of Smoluchowsky was modified to Smoluchowski.

-review the document especially the introduction, this should be more objective and brief, include the importance of this technique and its possible application

In the Introduction section, the importance of the technique is shown from lines 61 to 64. Our group considers that the extension of the introduction is adequate because the introduction addresses comparative aspects or experimental design with others in the literature.

-check the units used in the methodology section

Re: The units were corrected.

-please consider improving the wording of the methodology, it is incomplete and unclear, put the amount of monomers used followed by the rate as an indication

Re: The redaction of the methodology was modified.

-divide the characterization into sections

Re: The characterization was divided in sections

Improve figures and figure legends, they look incomplete or have errors

Re: These indications were realized

-include H-NMR spectra in the main text

Re: H-NMR spectra was collocated in main text
